# The relationship of tidal volume and driving pressure with mortality in hypoxic patients receiving mechanical ventilation

Robert A. Raschke[1]*, Brenda Stoffer[2], Seth Assar[3], Stephanie Fountain[3], Kurt Olsen[3], C. William Heise[1], Tyler Gallo[1], Angela Padilla-Jones[1,4], Richard Gerkin[1,5], Sairam Parthasarathy[6], Steven C. Curry[1,4]

1 The Division of Clinical Data Analytics and Decision Support, Department of Medicine, University of Arizona College of Medicine-Phoenix, Phoenix, AZ, United States of America, 2 Information Technology, Banner Health, Phoenix, AZ, United States of America, 3 Pulmonary Critical Care Medicine Fellowship, University of Arizona College of Medicine-Phoenix, Phoenix, AZ, United States of America, 4 Department of Medical Toxicology, Banner—University Medical Center Phoenix, Phoenix, AZ, United States of America, 5 Department of Medicine, Banner—University Medical Center—Phoenix, Phoenix, AZ, United States of America, 6 Division of Pulmonary, Allergy, Critical Care and Sleep Medicine, Department of Medicine University of Arizona College of Medicine, Tucson, AZ, United States of America

* Raschkebob@gmail.com

**Data Availability Statement:** All relevant data are within the manuscript and its Supporting Information files.

## Abstract

### Purpose

To determine whether tidal volume/predicted body weight (TV/PBW) or driving pressure (DP) are associated with mortality in a heterogeneous population of hypoxic mechanically ventilated patients.

### Methods

A retrospective cohort study involving 18 intensive care units included consecutive patients ≥18 years old, receiving mechanical ventilation for ≥3 days, with a $PaO_2/FiO_2$ ratio ≤300 mmHg, whether or not they met full criteria for ARDS. The main outcome was hospital mortality. Multiple logistic regression (MLR) incorporated TV/PBW, DP, and potential confounders including age, APACHE IVa® predicted hospital mortality, respiratory system compliance ($C_{RS}$), and $PaO_2/FiO_2$. Predetermined strata of TV/PBW were compared using MLR.

### Results

Our cohort comprised 5,167 patients with mean age 61.9 years, APACHE IVa® score 79.3, $PaO_2/FiO_2$ 166 mmHg and $C_{RS}$ 40.5 ml/cm $H_2O$. Regression analysis revealed that patients receiving DP one standard deviation above the mean or higher (≥19 cmH_20) had an adjusted odds ratio for mortality ($OR_{mort}$) = 1.10 (95% CI: 1.06–1.13, p = 0.009). Regression analysis showed a U-shaped relationship between strata of TV/PBW and adjusted mortality. Using TV/PBW 4–6 ml/kg as the referent group, patients receiving >10 ml/kg had similar adjusted $OR_{mort}$, but those receiving 6–7, 7–8 and 8–10 ml/kg had lower adjusted $OR_{mort}$

**Funding:** This study was funded in part by grant 2196 from the Flinn Foundation. SP was funded by NIH(HL126140, HL151254, AG059202, AI135108, HL140144, HL128954) and PCORI (DI- 2018C2-13161, EADI-16493, CER-2018C2-13262. The funders had no role in study design, data collection and analysis, decision to publish, or preparation of the manuscript.

**Competing interests:** The authors have declared that no competing interests exist.

**Abbreviations:** APACHE, Acute Physiology and Chronic Health Evaluation; ARDS, Acute respiratory distress syndrome; CDS, Clinical decision support; $C_{RS}$, Compliance of respiratory system; DP, Driving pressure; EMR, Electronic medical record; $FiO_2$, Fraction of inspired oxygen; LTVV, Low tidal volume ventilation; ITVV, Intermediate tidal volume ventilation; MLR, Multiple logistic regression; $OR_{mort}$, Odds ratio for mortality; $PaCO_2$, Partial pressure of arterial blood carbon dioxide; $PaO_2$, Partial pressure of arterial blood oxygen; PBW, Predicted body weight; PEEP, Positive end-expiratory pressure; $P_{PLAT}$, Plateau pressure; SD, Standard deviation; TV, Tidal volume.

(95%CI) of 0.81 (0.65–1.00), 0.78 (0.63–0.97) and 0.80 0.67–1.01) respectively. The adjusted $OR_{mort}$ in patients receiving 4–6 ml/kg was 1.26 (95%CI: 1.04–1.52) compared to patients receiving 6–10 ml/kg.

## Conclusions

Driving pressures ≥19 $cmH_2O$ were associated with increased adjusted mortality. TV/PBW 4-6ml/kg were used in less than 15% of patients and associated with *increased* adjusted mortality compared to TV/PBW 6–10 ml/kg used in 82% of patients. Prospective clinical trials are needed to prove whether limiting DP or the use of TV/PBW 6–10 ml/kg versus 4–6 ml/kg benefits mortality.

## Introduction

Previous literature suggests that modifying the ventilator parameters, tidal volume/predicted body weight (TV/PBW) and/or driving pressure (DP), improves survival from acute respiratory distress syndrome (ARDS). However, methodological shortcomings of that literature and limitations of the diagnostic criteria for ARDS described below prevent straightforward translation of these results into population-based quality improvement.

The optimal specific approach to titrating ventilator parameters remains unclear. The most widely-recommended approach, low tidal volume ventilation (LTVV) specifically targets TV/PBW 4–6 ml/kg if tolerated by the patient, but may range up to 8 ml/kg [1, 2]. Clinical trials that support the use of LTVV largely focused on patients with ARDS and compared LTVV with high control tidal volumes ranging 10–15 ml/kg [1–8]. Tidal volumes this high are rarely clinically employed [9], and few data are available to compare LTVV with *intermediate* tidal volumes (ITVV) such as in the range of 8–10 ml/kg commonly used in current practice.

A meta-analysis of LTVV clinical trials suggested that *driving pressure* (DP), rather than TV/PBW, is the modifiable ventilator parameter independently associated with mortality in patients with ARDS [10], but prospective trials of DP-limiting ventilator management have not been performed. A clinical practice guideline endorsed by multiple international professional societies recommends LTVV for patients with ARDS [2]. Some have recommend LTVV for virtually all mechanically ventilated patients [11–13]. Others recommend a DP-limiting ventilator strategy [14, 15].

Much of the previous literature on modifiable ventilator parameters focused on patients with ARDS. However, the diagnosis of ARDS may be difficult to operationalize in clinical practice. ARDS is conceptually a clinical-*pathological* entity [16] diagnosed using clinical-*radiological* criteria [17] for which clinicians and researchers have demonstrated limited inter-rater reliability and accuracy [18–22]. Furthermore, current data suggest that limiting TV/PBW or DP may also benefit patients *without* ARDS [11–13, 15, 23–25], suggesting that making the diagnosis of ARDS is not essential to deciding whether to limit TV/PBW or DP. In fact, an observational study including 459 ICUs in 50 countries showed that after adjusting for potentially confounding variables, clinical recognition of ARDS did not significantly influence the choice of TV/PBW, or whether DP was measured [9]. Population-based quality improvement efforts would be greatly simplified if the necessity to distinguish patients with ARDS from others with similar hypoxic ventilatory failure could be circumvented.

Our goal has been to implement evidence-based ventilator practice in a large healthcare system to improve survival of mechanically ventilated patients. But in order to do so, we needed

to better understand the relationship between modifiable ventilator parameters and mortality in a heterogeneous population of hypoxic mechanically ventilated patients. The specific aims of our study were: 1) To determine whether DP, TV/PBW, or subcategories of TV/PBW (4–6, 6–7, 7–8, 8–10, and >10 ml/kg) were independently associated with adjusted hospital mortality in hypoxic adult patients receiving ≥3 days of mechanical ventilation, regardless of whether they met ARDS diagnostic criteria, and 2) to establish discrete thresholds for optimal TV/PBW and/or DP.

## Methods

### Study design

This retrospective cohort study was performed as part of a quality improvement project, and was determined by our institution's Research Determination Committee to not require Institutional Review Board review. The study setting included the medical/surgical, cardiovascular, neurological, transplant and trauma intensive care units (ICUs) of 18 acute care hospitals within a large healthcare system in the southwestern United States between February 1, 2017 and January 31, 2019.

### Participants

Consecutive patients were included based on the inclusion criteria: ≥18 years of age, received volume control mechanical ventilation for at least three days, had a $PaO_2/FiO_2$ ratio ≤300 mmHg while receiving PEEP ≥5 $cmH_2O$ during the first 24 hours of mechanical ventilation, met criteria for calculation of an APACHE IVa score, had height recorded (with which to calculate PBW). Patients with less than three ventilator days were excluded in order to focus the analysis on patients more likely to accrue lung injury from prolonged mechanical ventilation. All patients were followed-up until hospital discharge or death.

### Variables

The main outcome variable was hospital mortality. The two predictor variables of interest were the modifiable ventilator parameters: TV/PBW and DP. Potential confounding variables included age, $PaO_2/FiO_2$ ratio, $PaCO_2$, respiratory system compliance ($C_{RS}$), APACHE IVa® predicted hospital mortality, hospital site, and the annual quarter in which the patient was admitted.

### Data sources

We used previously described bioinformatics [26, 27] embedded within a Cerner Millenium® electronic medical record (EMR) and an honest broker system to collect de-identified clinical and ventilator parameters on all patients receiving mechanical ventilation. These data included: age, gender, height (cm), $PaO_2$ (mmHg), $PaCO_2$ (mmHg), $FiO_2$, positive end-expiratory pressure (PEEP $cmH_2O$), set tidal volume (ml), and plateau pressure ($cmH_2O$). The first complete set of data for each patient obtained within 24 hours of intubation was used in the analysis described below. We used the APACHE IVa® severity scoring system (Cerner Corp, Kansas City MO) to enumerate hospital mortality, predicted hospital mortality and ventilator days.

## Study size

We calculated that 1248 patients were needed per TV/PBW stratum (for instance comparing patients receiving 4-6ml/kg to those receiving 8-10ml/kg) to provide 80% power to discern a 5% difference in mortality, assuming baseline mortality of approximately 25%.

## Statistical analysis

The following values were calculated for each patient: $PaO_2/FiO_2$, PBWmen = *50 + 0.91(cm of height—152.4)*; PBWwomen = *45.5 + 0.91(cm of height—152.4)*, TV/PBW, $C_{RS}$ = *TV/ ($P_{PLAT}$ -PEEP) (ml/cmH_2O)* and DP = *$TV/C_{RS}$ (cmH_2O)*.

The APACHE IVa[®] severity scoring system (Cerner Corp, Kansas City MO) incorporated chronic health conditions, 115 discrete admission diagnostic categories, and 27 clinical variables, including age, vital signs, Glasgow Coma Scale score, $FiO_2$, $PaO_2$, $PaCO_2$, arterial pH, urine output, creatinine, bilirubin, albumin, glucose, white blood cell count and hematocrit, with a reported a discriminant accuracy of 88% for predicting hospital mortality [28].

We performed *three* step-wise forward multiple logistic regression (MLR) analyses to investigate the association between TV/PBW, DP and hospital mortality with adjustment for confounders. The first MLR incorporated TV/PBW and all potential confounders listed above; the second included DP, TV/PBW and all confounders. Next we segregated patients into five strata of TV/PBW: (4–6, 6–7, 7–8, 8–10, and >10 mL/kg), chosen *a-priori* based on their relation to the design of multiple previous clinical trials [1–5, 11–13, 23, 29], and performed the third MLR forcing the five TV/PBW strata into the model as nominal variables, including DP and all potential confounders. We used this MLR to calculate the adjusted odds ratio for mortality ($OR_{mort}$) with 95% confidence intervals for each TV/PBW strata. We used the TV/PBW 4–6 ml/kg strata as the referent, based on the proposition that this strata (strict LTVV) theoretically represents best practice. In a *post-hoc* analysis, we combined three strata that had similar $OR_{mort}$ including tidal volumes in the 6–10 ml/kg range and used it as the referent to calculate the adjusted $OR_{mort}$ in the 4–6 ml/kg stratum.

We used the significant $OR_{mort}$ for a 1 standard deviation (SD) increase in DP to calculate a threshold DP (mean DP + 1 SD) above which adjusted mortality was significantly increased.

## Post-hoc MARS and sensitivity analysis

When the relationship between TV/PBW and mortality was observed to be non-linear based on the first MLR analysis described above, we performed *post-hoc* multivariate adaptive regression splines (MARS) analysis of the relationship between TV/PBW and adjusted mortality.

We used STATA[®] Version 15 (Statacorp, College Station, TX) for all statistical analyses.

## Results

Our CDS system identified 21,851 discrete episodes of mechanical ventilation received by 20,703 patients during the 2-year study period. 20,057 (96.9%) of these patients received volume control mechanical ventilation, and 14,320 (72.4%) had a $PaO_2/FiO_2$ ratio ≤300 mmHg during the first 24 hours of mechanical ventilation. 5,658 (39.5%) of patients with a $PaO_2/FiO_2$ ≤300 mmHg accrued ≥3 ventilator days. Three hundred and forty-seven (6.2%) of these did not have a height recorded, and 144 (2.6%) failed to satisfy criteria for APACHE IVa[®] mortality prediction and were excluded (**Fig 1**).

**Clinical characteristics** of the 5,167 study patients are shown in **Table 1**. The mean age was 61.9 years, and 42.4% were women. The most common admission diagnoses were pneumonia (30.0%), non-pulmonary sepsis (10.8%), cardiopulmonary arrest (10.2%), respiratory failure

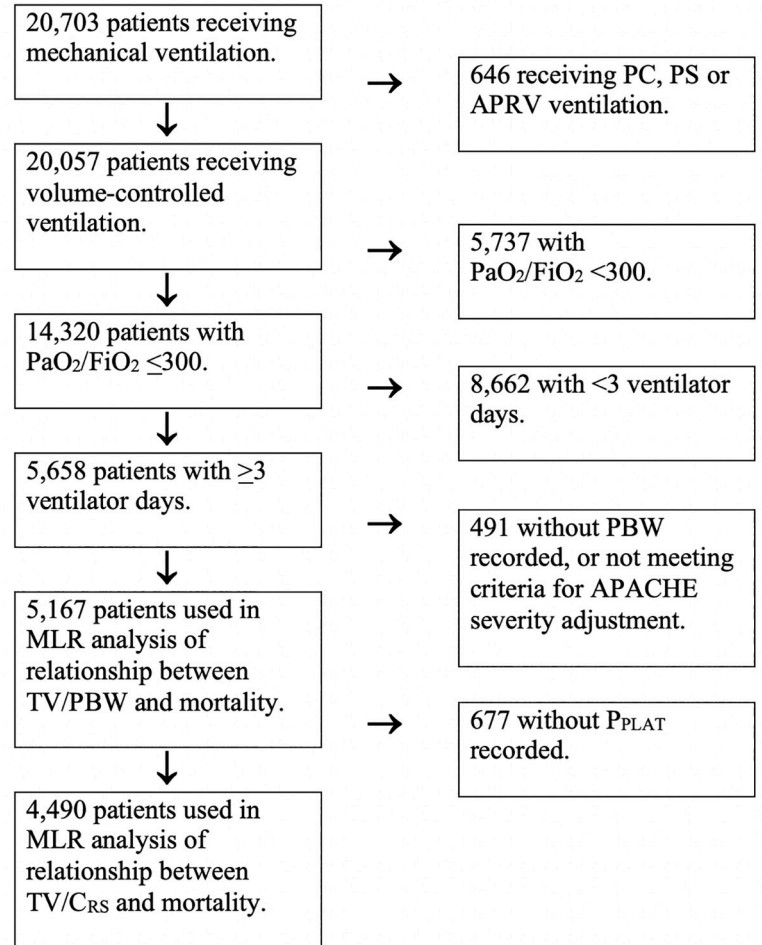

**Fig 1. Flow diagram of cohort inclusion/exclusion criteria.** [Abbreviations: APACHE: Acute physiology and chronic health evaluation, APRV: Airway pressure release ventilation, MLR: Multiple logistic regression, PC: Pressure control, $P_{PLAT}$: Plateau pressure, PS: Pressure support, DP: Driving pressure].

not due to pneumonia (8.8%), and non-cardiovascular surgery (8.7%). The mean APACHE IVa® score was 79.3, and mean $PaO_2/FiO_2$ ratio was 166 mmHg. The mean applied TV/PBW was 7.25 ml/kg. $P_{PLAT}$ was recorded in 4,490/5,167 (86.9%) patients, from which $C_{RS}$ and DP were calculated, yielding means of 20.7 cmH$_2$O, 40.5 ml/cmH$_2$O, and 13.7 cmH$_2$O, respectively. The median ventilator length of stay was five days (IQR: 4–9 days), and overall hospital mortality was 28.4% (95%CI: 27.1–29.6%). The distribution of TV/PBW received is shown in **Fig 2**.

**Table 2** shows the model resulting from the first MLR analysis of the relationship between TV/PBW and hospital mortality. Age, APACHE predicted hospital mortality, $PaO_2/FiO_2$, and admission to two particular hospitals out of the 18 participating in the study were significantly associated with mortality, but TV/PBW taken as a continuous variable, was not.

Post-hoc MARS analysis confirmed a significant U-shaped relationship between TV/PBW and adjusted mortality, which was down-sloping (slope -0.17, p = 0.001) as TV/PBW increased from 4 to 7.1 ml/kg, and up-sloping (slope 0.20, P = 0.001) as TV/PBW increased above 7.1 ml/kg.

**Table 1. Characteristics of 5,167 study patients.**

| | Mean | SD |
|---|---|---|
| | (unless otherwise noted) | (unless otherwise noted) |
| **Female gender** | 42.4% | |
| **Age (years)** | 61.9 | 16.9 |
| **PBW (kg)** | 64.3 | 10.9 |
| **APACHE IVa® score** | 79.3 | 28.0 |
| **ICU admission diagnosis categories:** | N | |
| Pneumonia with or without sepsis | 1548 (30.0%) | |
| Sepsis (non-pulmonary) | 559 (10.8%) | |
| Cardiorespiratory arrest | 526 (10.2%) | |
| Pulmonary (not pneumonia) | 453 (8.8%) | |
| Surgery (non-cardiovascular) | 448 (8.7%) | |
| Cardiology | 345 (6.7%) | |
| Neurology | 343 (6.6%) | |
| Surgery (cardiovascular) | 321 (6.2%) | |
| Trauma | 238 (4.6%) | |
| Toxic/metabolic | 176 (3.4) | |
| Gastrointestinal | 137 (2.7%) | |
| All other | 73 (1.4%) | |
| **Pulmonary parameters:** | | |
| $PaO_2/FiO_2$ (mmHg) | 165.6 | 70.0 |
| $C_{RS}$ (ml/cmH$_2$O) | 40.5 | 32.1 |
| **Ventilator settings/parameters** | | |
| Tidal volume (ml) | 457.4 | 74.9 |
| TV/PBW (ml/kg) | 7.25 | 1.34 |
| PEEP (cmH$_2$O) | 7.0 | 3.0 |
| | 5.0 (median) | IQR: 5–10 |
| $P_{PLAT}$ (cmH$_2$O) | 20.7 | 6.3 |
| DP (cmH$_2$O) | 13.7 | 5.4 |
| **Outcomes** | | |
| Hospital mortality | 28.4% | |
| APACHE IVa® Obs/Exp mortality | 0.980 | |
| Ventilator days | 5 (median) | IQR: 4–9 |

**Table 3** shows the results of second MLR analysis, of the relationship between DP, TV/PBW and hospital mortality. DP was found to be significantly associated with mortality after adjustment for significant confounders with an $OR_{mort}$ of 1.10 (95% CI: 1.06–1.13; p = 0.009) for each one standard deviation (SD) increase in DP. Seven hundred twenty-six patients received DP one standard deviation or greater above the mean (≥19 cmH$_2$O), with an $OR_{mort}$ of 1.15 (95%CI: 1.01–1.30), representing an estimated 33 deaths attributable to excessive DP. Age, APACHE predicted hospital mortality, $PaO_2/FiO_2$, and admission to two individual hospitals were also significant in this MLR model, which had Nagelkerke's pseudo-$R^2$ of 0.143.

**Table 4** shows the results of third MLR analysis of the relationship between five strata of TV/PBW, DP and all significant confounders. Patients receiving 4–6 ml/kg were used as a referent for comparison with the other groups–ie. the OR for each of the other strata were compared to the 4–6 ml/kg strata. Patients receiving 6–7 ml/kg and 7–8 ml/kg had significantly lower adjusted $OR_{mort}$ (0.81 p = 0.05, and 0.78 p = 0.03 respectively. Patient's receiving 8–10 ml/kg had adjusted $OR_{mort}$ of 0.80 with p = 0.06. The stratified analysis results from **Table 4**

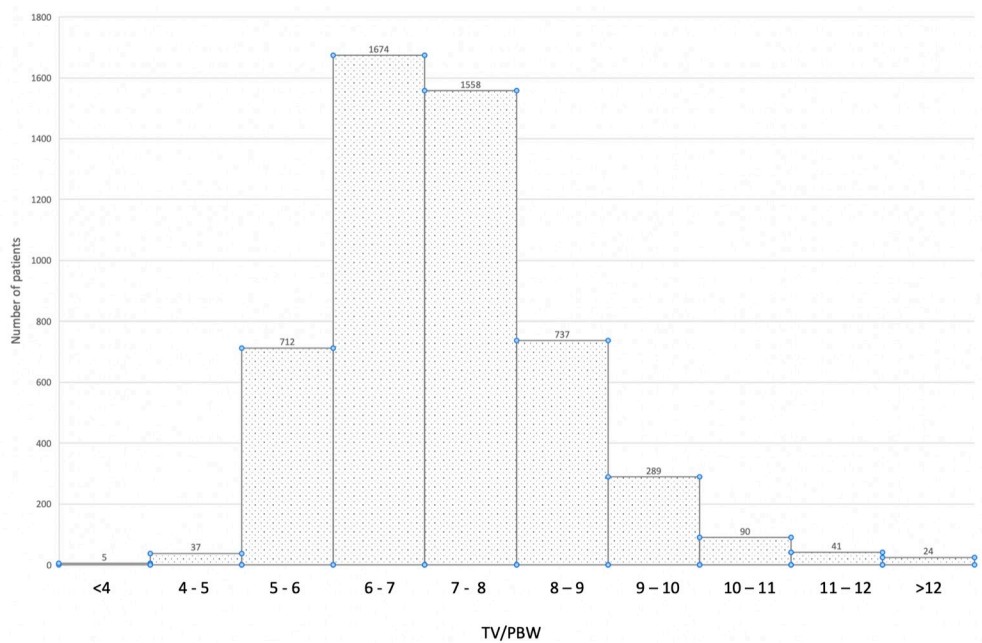

**Fig 2. Distribution of TV/PBV received by 5,167 cohort patients.**

are illustrated in **Fig 3**. Since the adjusted $OR_{mort}$ for TV 6–7, 7–8 and 8–10 ml/kg were all similar, we combined them together in *post-hoc* analysis, and using *them* as the referent group, the comparative adjusted $OR_{mort}$ for patients receiving 4–6 ml/kg was 1.26 (95% CI: 1.04–1.52; p = 0.02).

## Discussion

We report a large observational cohort study designed to examine the relationship between modifiable ventilator parameters and hospital mortality in hypoxic patients receiving mechanical ventilation. Our study cohort comprised about 25% of all patients receiving mechanical ventilation in our healthcare system, with a crude mortality of 28%, comparable to that of mild ARDS [17, 30]. Our patients were also similar to patients with ARDS in terms of age, $PaO_2/FiO_2$, $C_{RS}$, TV and $P_{PLAT}$ [9, 30]. A recent epidemiologic study suggests that more than 50% receiving mechanical ventilation with hypoxic respiratory failure, such as those included in our study, meet Berlin criteria for ARDS [30].

**Table 2.**

| Significant independent variables in the first MLR model: | Odds ratio* (95% CI) | P value |
|---|---|---|
| **Age** | 1.16 (1.09–1.24) | <0.001 |
| **APACHE predicted mortality** | 1.99 (1.86–2.14) | <0.001 |
| **$PaO_2/FiO_2$** | 0.87 (0.81–0.94) | <0.001 |
| **Admission to hospital X** | 1.64 (1.32–2.04) | <0.001 |
| **Admission to hospital Y** | 0.56 (0.36–0.86) | 0.008 |

*Odds ratios are associated with a one standard deviation (SD) increment in the given variable, except in the case of admission to hospitals X or Y. Values used for SD: Age, 16.9 years; risk of death 26%; $PaO_2/FiO_2$, 70.

**Table 3.**

| Significant independent variables in the second MLR model: | Odds ratio* (95% CI) | P value |
|---|---|---|
| **Driving pressure** | 1.10 (1.06–1.13) | 0.009 |
| **Age** | 1.17 (1.09–1.26) | <0.001 |
| **APACHE predicted mortality** | 1.98 (1.90–2.05) | <0.001 |
| **PaO$_2$/FiO$_2$** | 0.87 (0.80–0.94) | 0.002 |
| **Admission to hospital X** | 1.74 (1.54–1.95) | <0.001 |
| **Admission to hospital Y** | 0.53 (0.44–0.68) | 0.006 |

*Odds ratios are associated with a one standard deviation (SD) increment in the given variable, except in the case of admission to hospitals X or Y. Values used for SD: Age, 16.9 years; risk of death 26%; PaO$_2$/FiO$_2$, 70; DP, 5.42 cmH$_2$O.

We adopted a pragmatic approach to patient selection, breaking with the convention of using ARDS as a selection criteria. ARDS is conceptually a clinical-*pathological* entity [16], but is currently diagnosed using imperfect clinical-*radiological* criteria [17] for which clinicians have demonstrated poor accuracy and low inter-rater reliability [18, 19]. Previous studies have shown that even researchers with expertise in ARDS have only moderate agreement when applying the clinical diagnosis of ARDS [20–22]. In contrast, the PaO$_2$/FiO$_2$ ratio criteria we used to select our study patients was easy to accurately extract from the EMR and represents the only ARDS criteria independently associated with mortality [17]. This approach to patient selection makes our study unique in the context of related literature and eases translation of our results into population-based quality improvement.

In our first two multivariate analyses, we found that DP rather than TV/PBW was the modifiable ventilator parameter independently associated with survival. But our *stratified* analysis and multivariate adaptive regression splines analysis showed a significant U-shaped relationship between TV/PBW and adjusted mortality. Patient receiving 4–6 ml/kg had similar mortality to those receiving >10 ml/kg. But patients receiving 6–7, 7–8 and 8–10 ml/kg all had comparative adjusted OR$_{mort}$ about 0.80. These findings raise two provocative hypotheses. The current range of LTVV (4–8 ml/kg) might be composed of substrata with distinctly different mortality effects: TV in the range of 6-8ml/kg may be safer than 4–6 ml/kg. Intermediate tidal volume ventilation (ITVV) might not be inferior to LTVV. Disregarding previous convention, we might hypothesize the optimal TV/PBW range to be 6–10 ml/kg based on these observational findings. Using TV 6–10 ml/kg as the referent group, the comparative adjusted OR$_{mort}$ for patients receiving 4–6 ml/kg, such as targeted in the ARMA study, is 1.26 (95% CI: 1.04–1.52; p = 0.02).

A U-shaped relationship between TV/PBW and mortality was first posited by Eichacker and colleagues in their 2002 meta-analysis of ARDS trials testing LTVV [31], and was later described in a cohort of 5,183 mechanically ventilated patients that found that those receiving ≤6 ml/kg and those receiving >10 ml/kg had increased mortality (OR 1.23 and 1.14 respectively) compared to patients receiving 6–10 ml/kg (p = 0.09) [32]. A U-shaped relationship was not demonstrated in a much smaller cohort study (N = 485) that was not specifically powered to do so [33].

A U-shaped relationship between TV/PBW and mortality has sound physiological explanation [31]. Tidal volumes that are too high lead to alveolar overdistention, ventilator-induced lung injury, systemic inflammation, ventilator non-triggering, and eccentric respiratory muscle injury [1, 34–37]. Tidal volumes that are too low can cause hypercarbic acidosis, increased work of breathing, and patient-ventilator dyssynchrony [38–40]. The later can manifest as

**Table 4. Adjusted odds ratio for mortality in five strata of TV/PBW.**

|  | Strata of TV/PBW (ml/kg) | | | | |
| --- | --- | --- | --- | --- | --- |
|  | **4–6** | **6–7** | **7–8** | **8–10** | **>10** |
| **N** | 760 | 1672 | 1554 | 1026 | 155 |
| **Median TV/PBW** | 5.8 | 6.5 | 7.5 | 8.6 | 11.0 |
| **Multivariate $OR_{mort}$ mortality (95% CI)** | 1.00* | 0.81 | 0.78 | 0.80 | 1.03 |
|  |  | (0.65–1.00) | (0.63–0.97) | (0.63–1.01) | (0.67–1.59) |
| **P-value from MLR (comparing adjusted $OR_{mort}$ to that of the 4–6 ml/kg)** |  | 0.05** | 0.03** | 0.06 | 0.90 |

Abbreviations. TV/PBW: Tidal volume/predicted body weight, $OR_{mort}$: Odds ratio for mortality, CI.: Confidence interval

*TV/PBW 4-6ml/kg is the referent group for the OR analysis.

** Patients receiving 6–7 and 7–8 ml/kg had significantly lower adjusted mortality than those receiving 4–6 ml/kg TV/PBW.

strenuous inspiratory efforts and double-triggering, either of which can paradoxically lead to alveolar overdistention [40–43]. Insufficient TV/PBW can also cause atelectrauma, increased respiratory rate (stress frequency) and increased sedation requirements [44]. The use of LTVV in the subset of ARDS patients with relatively preserved $C_{RS}$ has been shown to be associated with increased mortality [45].

Current evidence supporting LTVV has several important limitations. Clinical trials used to support LTVV [1, 3–8] did not perform comparative analysis of TV/PBW substrata comprising LTVV, and used tidal volumes (10–15 ml/kg PBW) in their control groups that were significantly higher than the intermediate tidal volumes commonly used in clinical practice at the time [31, 32, 46, 47]. Several authors have posited that the apparent benefit of LTVV in these trials might be solely attributable to the injuriously high tidal volumes and $P_{PLAT}$ received by control patients rather than to any specific benefit of LTVV [31, 48]. Deans and colleagues analyzed 2,587 patients who met enrollment criteria for the landmark ARMA trial [1], but

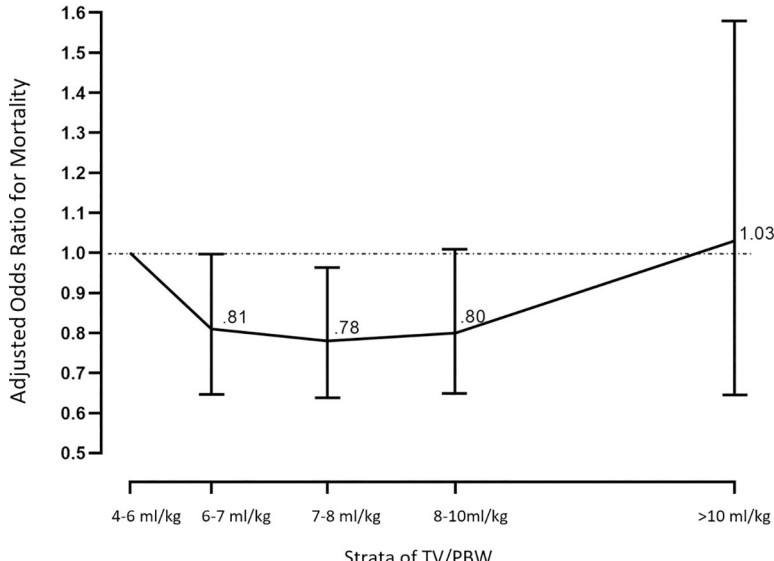

**Fig 3. Adjusted odds ratio for mortality for five strata of TV/PBW (4–6, 6–7, 7–8, 8–10 and >10 ml/kg).** The median TV/PBW for each strata is plotted on the X-axis. Adjusted $OR_{mort}$ with 95% CI error bars plotted on the Y-axis for the second through fifth strata. [The first stratum (4–6 ml/kg) is the referent for the calculation of OR for each of the other strata].

were excluded for technical reasons such as lack of consent; these patients went on to receive conventional tidal volume management, yet achieved mortality of 31.7%—comparable to the 31.0% mortality of study patients that received LTVV and significantly *lower* than the 39.8% mortality experienced by study patients randomized to the control group [45]. In 2017, an international consensus of critical care societies published the results of a meta-analysis of nine randomized controlled trials comparing LTVV (4–8 ml/kg PBW) to "traditional" tidal volumes (10–15 ml/kg PBW) in patients with ARDS. They found *no* difference in mortality (risk ratio: 0.87–95% CI: 0.70–1.08) in their primary analysis [2] and provided no data comparing LTVV to ITVV, yet "strongly recommended" LTVV. Less than 5% of patients in a large epidemiological study of ARDS [9] and less than 3% of patients in our cohort received tidal volumes in the range >10 ml/kg used in the control groups of the studies included in this meta-analysis, so it's relevance to current practice would be questionable even if it's findings had been significant.

As early as 2002, Eichacker posited that ITVV (with limited $P_{PLAT}$) was not inferior, and might be *superior* to LTVV, based on a patient data-level meta-analysis of ARDS clinical trials [31]. The only previous randomized controlled trial we are aware of that specifically compared LTVV to ITVV included 961 patients *without* ARDS, and showed no difference in mortality, ventilator days, length of stay, or pulmonary complications [29].

Although we cannot calculate the proportion of our cohort with ARDS, it is noteworthy that 20 years after the ARMA trial supporting LTVV 4-6ml/kg [1], only 14.7% of our patients received it, compared to the majority 82% receiving 6–10 ml/kg. Several previous reports similarly show that although most clinicians agree that LTVV should be used in patients with ARDS, only 7–19% of their ARDS patients receive it [49, 50]. An observational study including 459 ICUs in 50 countries showed that tidal volumes of 6–10 ml/kg were used in approximately 75% of ARDS patients, four times more often than 4-6ml/kg was used, and that clinical recognition of ARDS did not significantly influence the TV/PBW administered [9]. Our study supports this practical approach to ventilator management, even though it sometimes incorporates non-recommended ITVV. It is possible that clinicians are *correctly* choosing higher range LTVV and ITVV ventilation to avoid complications of strict 4–6 mg/kg LTVV.

Our observation that DP is related to mortality supports and extends the findings of a prior clinical trial meta-analysis [10], by showing that the relationship between DP and mortality holds in a heterogenous group of hypoxic patients, regardless of whether they have ARDS, even though clinical measurement of $C_{RS}$ by respiratory care practitioners in the presence of spontaneous inspiratory efforts could introduce significant error into the measurement or calculation of $P_{PLAT}$, $C_{RS}$ and DP [14]. The later concern has previously been posited as an unresolved barrier to clinical implementation of DP-limited mechanical ventilation [42]. It is worth noting that DP equals TV/$C_{RS}$–therefore titration of DP can be seen as a form of precision medicine in which TV is matched to an individual patient's respiratory system compliance.

Several studies provide context for our finding of a relationship between DP and mortality in hypoxic ventilated patients. A meta-analysis of clinical trials and observational studies [51] and three subsequent cohort studies [15, 30, 52] showed a significant relationship between DP and mortality in patients with ARDS. In patients *without* ARDS, a quasi-experimental trial [25], and two cohort studies [15, 24] showed that DP was associated with mortality. Of note, the latter study showed a significant relationship only in non-ARDS patients with $PaO_2/FiO_2$ <300 mmHg. One recent observational study that did *not* require hypoxia as an inclusion criterion showed no relationship between DP and mortality in non-ARDS patients [52]. Reported threshold values for DP associated with increased mortality are in a relatively narrow range of 14–19 cmH$_2$O [10, 30, 53], consistent with our results. Taken together, considerable data

support the contention that reducing DP <19 cmH$_2$O would make a reasonable target to reduce mortality in mechanically ventilated patients.

## Limitations

Residual bias/confounding remains a major threat to the validity of our observational study. The heterogeneity of our cohort and the real-world shortcomings of clinical data used in our study reduce internal validity, but increase the translational utility of our findings. Our exclusion of patients with less than three ventilator days could have introduced bias if such patients are especially prone to ventilator induced lung injury. We used only a single set of ventilator parameters per patient to describe ventilator management–a practical decision based on the large amount of clinical data we accessed. Our simplified calculation of C$_{RS}$ did not take into account spontaneous patient effort. Elevated DP could be a marker of more severe underlying lung injury rather than a *cause* of increased mortality, notwithstanding our statistical adjustment for C$_{RS}$, PaO$_2$/FiO$_2$ and other confounders. We used APACHE IVa$^®$ predicted hospital mortality as an independent variable in our MLR, which has reduced discriminant accuracy in mechanically ventilated patients [54]. Although our MLR model was superior to APACHE IVa$^®$, it had only a modest pseudo-R$^2$ for predicting mortality. We therefore suspect that lowering DP may have only a modest effect on mortality of mechanically ventilated patients, in whom myriad other factors likely influence survival. Prospective clinical trials are needed to further investigate whether ITVV is superior to strict LTVV, and to demonstrate clinical efficacy of DP-limiting ventilator strategies.

## Conclusions

Driving pressures ≥19 cmH$_2$O were associated with increased adjusted hospital mortality in our retrospective cohort. We intend to use this finding to inform development of clinical decision support focused on limiting DP rather than achieving strict LTVV in our healthcare system. Tidal volumes of 4–6 ml/kg were used in less than 15% of patients and were associated with increased adjusted mortality. Tidal volumes of 6–10 ml/kg were used in 82% of patients and had significantly reduced adjusted mortality compared to that associated with tidal volumes 4–6 ml/kg. We hypothesize that ITVV may not be inferior to strict LTVV in patients with hypoxic respiratory failure.

## Supporting information

**S1 File.**
(XLSX)

## Acknowledgments

We acknowledge and appreciate the technical assistance of Nick Ernzen from Banner Health Information Technology, and biostatistical support from Dr. Paul Kang from the Biostatistics and Study Design Core at the University of Arizona College of Medicine–Phoenix, and Pooja Rangan, MBBS MPH from the Department of Internal Medicine, Banner University Medical Center—Phoenix.

## Author Contributions

**Conceptualization:** Robert A. Raschke, Seth Assar, C. William Heise, Tyler Gallo, Sairam Parthasarathy, Steven C. Curry.

**Data curation:** Robert A. Raschke, Brenda Stoffer, C. William Heise, Angela Padilla-Jones, Richard Gerkin.

**Formal analysis:** Robert A. Raschke, Brenda Stoffer, Angela Padilla-Jones, Richard Gerkin, Sairam Parthasarathy.

**Funding acquisition:** C. William Heise, Steven C. Curry.

**Investigation:** Robert A. Raschke, Seth Assar, Stephanie Fountain, Kurt Olsen, C. William Heise, Tyler Gallo, Richard Gerkin, Sairam Parthasarathy, Steven C. Curry.

**Methodology:** Robert A. Raschke, Richard Gerkin, Sairam Parthasarathy, Steven C. Curry.

**Project administration:** Angela Padilla-Jones, Steven C. Curry.

**Resources:** Robert A. Raschke, Brenda Stoffer, Seth Assar, Stephanie Fountain, Kurt Olsen, Angela Padilla-Jones, Richard Gerkin, Steven C. Curry.

**Software:** Robert A. Raschke, Brenda Stoffer, Richard Gerkin, Steven C. Curry.

**Supervision:** Robert A. Raschke, Steven C. Curry.

**Validation:** Robert A. Raschke, Richard Gerkin.

**Visualization:** Robert A. Raschke, Sairam Parthasarathy, Steven C. Curry.

**Writing – original draft:** Robert A. Raschke.

**Writing – review & editing:** Robert A. Raschke, Brenda Stoffer, Seth Assar, Stephanie Fountain, Kurt Olsen, C. William Heise, Tyler Gallo, Angela Padilla-Jones, Richard Gerkin, Sairam Parthasarathy, Steven C. Curry.

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
