## [Decision Letter · Decision Letter 0]

14 Jun 2021

PONE-D-21-17095

The Relationship of Tidal Volume and Driving Pressure with Mortality in Hypoxic Patients Receiving Mechanical Ventilation.

PLOS ONE

Dear Dr. Raschke,

Thank you for submitting your manuscript to PLOS ONE. After careful consideration, we feel that it has merit but does not fully meet PLOS ONE’s publication criteria as it currently stands. Therefore, we invite you to submit a revised version of the manuscript that addresses the points raised during the review process.

We look forward to receiving your revised manuscript.

Kind regards,

Corstiaan den Uil

Academic Editor

PLOS ONE

Journal Requirements:

"This study was funded in part by grant 2196 from the Flinn Foundation. SP was funded by NIH

(HL126140, HL151254, AG059202, AI135108, HL140144, HL128954) and PCORI (DI-

2018C2-13161, EADI-16493, CER-2018C2-13262) at the time of writing this manuscript."

"No. The funders had no role in study design, data collection and analysis, decision to publish, or preparation of the manuscript."

Reviewers' comments:

Reviewer's Responses to Questions

**Comments to the Author**

1. Is the manuscript technically sound, and do the data support the conclusions?

Reviewer #1: Yes

Reviewer #2: Partly

2. Has the statistical analysis been performed appropriately and rigorously? 

Reviewer #1: Yes

Reviewer #2: I Don't Know

3. Have the authors made all data underlying the findings in their manuscript fully available?

Reviewer #1: Yes

Reviewer #2: No

4. Is the manuscript presented in an intelligible fashion and written in standard English?

Reviewer #1: Yes

Reviewer #2: Yes

5. Review Comments to the Author

Reviewer #1: In a retrospective review of over 5000 patients and involving 18 intensive care units, the authors establish the relationship between two mechanical ventilation parameters: driving pressure (DP) and tidal volume/predicted body weight (TV/PBW) and mortality. The study concluded that: (1) driving pressure is significantly associated with mortality, consistent with the findings of Amato et al, and therefore needs to be considered when titrating ventilator settings and (2) that low-tidal volume ventilation (LTVV) may not be the optimal mechanical ventilation strategy for all patients but rather that intermediate-tidal volume ventilation (ITVV) may lead to superior patient outcomes. I applaud the authors on their study and in challenging the concept of LTVV, which has long-deserved additional reflection given the inability to replicate the results of the initial ARMA study, and in the absence of longitudinal change in ARDS-related mortality.

Major Comments:

The authors demonstrate that both DP and VT/PBW significantly modify mortality risk, however they did not provide data comparing DP to the (five) VT/PBW strata. This is important because if the (average) driving pressures were similar (but with different VT/PBW), this suggests that the compliance of the patients was different. This might suggest that the differences in mortality may have been modified by underlying patient lung disease severity/compliance rather than the VT/PBW.

Minor Comments:

- These study results suggest that the ARMA trial did not establish that 6mL/kg was optimal but that 12mL/kg was sub-optimal. In support of the findings of this study, it would be worth including a mention of the article published by Deans et al (Crit Care Med 2005 – PMID: 15891350) in the discussion as that study retrospectively analyzed the ARMA data and demonstrated that the mortality rates between the group of patients receiving 6mL/kg and those who were excluded from the study and receiving then-conventional ventilation (ITVV) had similar mortality rates. [They also demonstrated that the impact of tidal volumes on mortality was related to lung compliance or - in essence, but not in name - driving pressure.]

- Recommend avoiding the use of contractions

- Although ITVV was defined as an abbreviation (on the "Abbreviations" page - 3), its first use in the text is in the discussion (page 17). Recommend defining it in the text prior to its use.

Reviewer #2: In this retrospective cohort, Dr Raschke et al studied over five thousand patients mechanically ventilated with PF ratios below 300 mHg in 18 ICUs in the southwestern United States. They found that tidal volume had a U-shaped relationship with hospital mortality and that driving pressure was also associated with hospital mortality even after adjusting for disease severity. I have some concerns and suggestions as detailed below.

1) My major concern is that the increased mortality in low tidal volume (<6mL/Kg PBW) be due to residual confounding. For example, patients with higher PEEP values due to hypoxemia might end up receiving lower tidal volumes to limit plateau pressure to less than 30 cmH2O as per protective ventilation protocols. I suggest that the authors include PEEP in the multivariable adjustment.

2) Respiratory system compliance can be a marker of the underlying severity of lung disease, but also varies according to patient lung size. Normalizing compliance to PBW (in mL/cmH2O/Kg PBW) helps take into account patient size. I suggest that the authors use normalized compliance in their multivariable adjustment.

3) Sample size was computed to find differences between groups. What groups? Please explain.

4) Why did the authors use ANOVA to compare the different TV/PBW strata? I believe that adding the strata to the multivariable logistic regression (as dummy variables) would have been a more natural choice.

5) Please explain in further detail how the driving pressure threshold of 19 cmH2O was found.

6) Please add the unit for the PF ratio (mmHg)

7) Is tidal volume still significantly associated with survival when driving pressure is in the model? This information is relevant because – if not – it would be simpler to target protection in terms of driving pressure.

8) Please discuss why the findings are in disagreement with those of Needham et al (BMJ 2012). In that prospective cohort study, they found that tidal volume had a linear relationship with survival. Modeling tidal volume with cubic splines did not improve the relationship.

9) How is it possible to reconcile the authors’ finding to those from Prevent trial (Simonis JAMA 2018) in which they tested a strategy with low vs intermediate tidal volumes in patients without ARDS?

6. PLOS authors have the option to publish the peer review history of their article (what does this mean?). If published, this will include your full peer review and any attached files.

Reviewer #1: **Yes: **Michaela Kollisch

Reviewer #2: No

---

## [Author Response · Author response to Decision Letter 0]

21 Jul 2021

Reviewer #1: In a retrospective review of over 5000 patients and involving 18 intensive care units, the authors establish the relationship between two mechanical ventilation parameters: driving pressure (DP) and tidal volume/predicted body weight (TV/PBW) and mortality. The study concluded that: (1) driving pressure is significantly associated with mortality, consistent with the findings of Amato et al, and therefore needs to be considered when titrating ventilator settings and (2) that low-tidal volume ventilation (LTVV) may not be the optimal mechanical ventilation strategy for all patients but rather that intermediate-tidal volume ventilation (ITVV) may lead to superior patient outcomes. I applaud the authors on their study and in challenging the concept of LTVV, which has long-deserved additional reflection given the inability to replicate the results of the initial ARMA study, and in the absence of longitudinal change in ARDS-related mortality.

Major Comments:

The authors demonstrate that both DP and VT/PBW significantly modify mortality risk, however they did not provide data comparing DP to the (five) VT/PBW strata. This is important because if the (average) driving pressures were similar (but with different VT/PBW), this suggests that the compliance of the patients was different. This might suggest that the differences in mortality may have been modified by underlying patient lung disease severity/compliance rather than the VT/PBW.

We agree with the reviewer and further note that TV/PBW is mathematically associated with driving pressure (DP= TV/CRS). Both are ways of expressing the TV, either in relation to the patients body size, or in relation to their respiratory compliance (CRS). This relationship was demonstrated by Amato and colleagues in their retrospective analysis of ARDSnet clinical trial data. 

We calculated DP for each of the five increasing strata of TV/PBW and they were positively associated as the reviewer suspected they would be: 12.8, 12.8, 13.7, 15.2, and 16.7 cmH2O respectively. However, compliance did not appear to be associated with increasing TV/PBW, with compliances of: 38.7, 43.3, 40.3, 37.4 and 40.7 ml/cmH2O respectively in the five strata of increasing TV/PBW. 

Compliance was adjusted for in all three of our MLRs and MARS analyses and therefore the associations we reported should not have been confounded by differences in compliance. We used the following additional covariates in our MLR to also adjust for severity of illness: PaO2/FiO2 ratio, PaCO2, and APACHE IVa predicted hospital mortality. But it is likely that these few variables do not eliminate confounding as suggested by the reviewer . We added the statement: “Residual bias/confounding remains a major threat to the validity of our observational study” to the discussion of our study limitations.

Minor Comments:

- These study results suggest that the ARMA trial did not establish that 6mL/kg was optimal but that 12mL/kg was sub-optimal. In support of the findings of this study, it would be worth including a mention of the article published by Deans et al (Crit Care Med 2005 – PMID: 15891350) in the discussion as that study retrospectively analyzed the ARMA data and demonstrated that the mortality rates between the group of patients receiving 6mL/kg and those who were excluded from the study and receiving then-conventional ventilation (ITVV) had similar mortality rates. [They also demonstrated that the impact of tidal volumes on mortality was related to lung compliance or - in essence, but not in name - driving pressure.]

The article by Deans was fascinating and we greatly appreciate the reference We have added it to support our discussion in several places, focusing on the equalivalent outcomes in study patients receiving LTVV and in patients excluded for technical reasons. We also thought the association Deans found between compliance and the response to LTVV could help explain the U-shaped relationship between TV/PBW and mortality that we describe and also added that to the discussion. 

- Recommend avoiding the use of contractions. Agreed and corrected throughout.

- Although ITVV was defined as an abbreviation (on the "Abbreviations" page - 3), its first use in the text is in the discussion (page 17). Recommend defining it in the text prior to its use. Agreed and corrected. ITVV is now defined in the introduction. 

Reviewer #2: In this retrospective cohort, Dr Raschke et al studied over five thousand patients mechanically ventilated with PF ratios below 300 mHg in 18 ICUs in the southwestern United States. They found that tidal volume had a U-shaped relationship with hospital mortality and that driving pressure was also associated with hospital mortality even after adjusting for disease severity. I have some concerns and suggestions as detailed below.

1) My major concern is that the increased mortality in low tidal volume (<6mL/Kg PBW) be due to residual confounding. For example, patients with higher PEEP values due to hypoxemia might end up receiving lower tidal volumes to limit plateau pressure to less than 30 cmH2O as per protective ventilation protocols. I suggest that the authors include PEEP in the multivariable adjustment.

We agree that confounding remains a major concern in our observational study. We selected potential confounders carefully when we developed the methods of our study. One factor in that consideration was that independent variables in the model should be mathematically independent of each other. Therefore we did not include plateau pressure or PEEP as independent variables, but instead used driving pressure (which was calculated as Pplat minus PEEP). 

In specific response to the reviewers concern, we calculated mean PEEP values in the five TV/PBW strata and they ranged from 6.2 to 7.9 mmHg – a difference of no more than 1.7mmHg between any two TV/PBW strata. Multiple clinical trials, including the ARDSnet trial, have failed to demonstrate a relationship between PEEP and mortality in patients with and without ARDS (1-3). Therefore, it seems unlikely that a difference in PEEP levels equal to, or less than 1.7 mmHg between groups would have a significant effect on the mortality in our study. We therefore respectfully declined reanalyzing our data based on this post-hoc consideration. It is our preference in general to stay true to the analysis we planned a-priori, while recognizing it’s limitations. 

We therefore added the statement: “Residual bias/confounding remains a major threat to the validity of our observational study findings” to the discussion of our study limitations.

1) The National Heart, Lung, and Blood Institute ARDS Clinical Trials Network. Higher versus Lower Positive End-Expiratory Pressures in Patients with the Acute Respiratory Distress Syndrome. N Engl J Med 2004; 351:327-33. DOI: 10.1056/NEJMoa032193

2) Writing Committee and Steering Committee for the RELAx Collaborative Group. Effect of a Lower vs Higher Positive End-Expiratory Pressure Strategy on Ventilator-Free Days in ICU Patients Without ARDS: A Randomized Clinical Trial. JAMA. 2020;324(24):2509–2520. DOI:10.1001/jama.2020.23517

3) Walkey AJ, Del Sorbo, L, Hodgson CL. Higher PEEP versus Lower PEEP Strategies for Patients with Acute Respiratory Distress Syndrome. A Systematic Review and Meta-Analysis. Annals American Thoracic Society 2017;17 Ann Am Thorac Soc Vol 14, Supplement 4, pp S297–S303 https://doi.org/10.1513/AnnalsATS.201704-338OT

2) Respiratory system compliance can be a marker of the underlying severity of lung disease, but also varies according to patient lung size. Normalizing compliance to PBW (in mL/cmH2O/Kg PBW) helps take into account patient size. I suggest that the authors use normalized compliance in their multivariable adjustment.

We considered this suggestion with caution since one of the aims of our study was to compare selecting TV based on PBW (TV/PBW) versus selecting TV based on compliance (CRS), since driving pressure = TV/CRS. Although CRS is not mathematically independent from driving pressure, we tried to avoid further mathematical-linking of variables in our model except when compelled. 

On review of landmark literature, we found that CRS not CRS/PBW, was used by the ARDS definition task force (4), the ARDSnet trial group (5), and the LUNG-SAFE investigators (6) among others, in their multivariate regression models. We do not find consensus in the literature that CRS/PBW is superior to CRS for such modeling. Again, we respectfully refer to our practice of restraint in reanalyzing data based on post-hoc considerations, and holding true to the analysis planned a-priori.

4) The ARDS Definition Task Force*. Acute Respiratory Distress Syndrome: The Berlin Definition. JAMA. 2012;307(23):2526–2533. doi:10.1001/jama.2012.5669

5) Amato, MBP, Meade MO, Slutsky AS, et al. Driving pressure and survival in the acute respiratory distress syndrome. N Engl J Med 2015; 372:747-755. DOI: 10.1056/NEJMsa1410639

6) Bellani G, Laffey JG, Pham T, et al. Epidemiology, Patterns of Care, and Mortality for Patients With Acute Respiratory Distress Syndrome in Intensive Care Units in 50 Countries. JAMA. 2016;315(8):788–800. doi:10.1001/jama.2016.0291

3) Sample size was computed to find differences between groups. What groups? Please explain.

We changed this section of methods in response to the reviewer’s suggestion: “We calculated that 1248 patients were needed per TV/PBW stratum (for instance comparing patients receiving TV/PBW <6ml/kg to those receiving 8-10ml/kg) to provide 80% power to discern a 5% difference in mortality, assuming baseline mortality of approximately 25%.”

4) Why did the authors use ANOVA to compare the different TV/PBW strata? I believe that adding the strata to the multivariable logistic regression (as dummy variables) would have been a more natural choice.

We appreciate the reviewer pointing out this statistical error. We performed the analysis as the reviewer recommended after reconsulting with our statistician, who agreed that the use of MLR was preferable to ANOVA. This analysis is identified in the methods section as the third MLR analysis. This caused our main statistical expression of mortality risk to change from observed/expected mortality to adjusted odds ratio for mortality. It changed the significance of some of our between-strata comparisons, although our conclusions remained generally robust. We subsequently had to redo table 4 and figure 3 using the new analysis, and adjust multiple parts of the paper, from the abstract through the conclusions. 

5) Please explain in further detail how the driving pressure threshold of 19 cmH2O was found.

We clarified this per the reviewer’s suggestion in the methods and results sections of the manuscript. We simply chose 1 standard deviation above the mean DP. The odds ratio for mortality for a 1 SD increase in DP was 1.10; 95% CI: 1.06 -1.13, p=0.009. 726/4490 of our cohort patients who had DP measured fell into this group, which seemed a reasonably-sized target group for clinical decision support. 

6) Please add the unit for the PF ratio (mmHg)

Done.

7) Is tidal volume still significantly associated with survival when driving pressure is in the model? This information is relevant because – if not – it would be simpler to target protection in terms of driving pressure.

TV was not significantly associated with survival in either of our first two MLR models – with or without driving pressure in the model. Therefore we concluded that we were going to limit driving pressure as our quality-improvement target. We state this in our conclusions: “We intend to use these findings to inform development of clinical decision support focusing on limiting DP rather than achieving strict LTVV in our healthcare system”. 

We did not expect or hypothesize a U-shaped relationship between TV/PBW and mortality a priori, and therefore our first two MLR analyses were not designed to detect such an association. We would likely have had to add-in quadratic transformations of TV/PBW in order to do so with MLR. But once we observed the U-shaped relationship, we felt it was worth further investigation. We performed a separate post-hoc MARS analysis, which statistically confirmed a significant U-shaped relationship. Thus, although TV/PBW was not a significant predictor in either of the first two MLR models (with or without DP), that doesn’t mean it’s not related to mortality. Our MLR models could not have detected a significant U-shaped relationship the way they were originally designed. It was only demonstrable when a post hoc statistical test designed to complex relationships was performed, and when TV/PBW strata were forced into the third MLR model as dummy variables (as you suggested). 

We don’t expect that our observational trial will appreciably change the widespread use of TV/PBW anytime soon, and we think that the hypothesis that intermediate TV may not be inferior to LTVV will remain clinically important and interesting to clinicians in the foreseeable future. We note in our discussion that a prospective clinical trials will be needed to confirm the hypotheses generated by our study. 

8) Please discuss why the findings are in disagreement with those of Needham et al (BMJ 2012). In that prospective cohort study, they found that tidal volume had a linear relationship with survival. Modeling tidal volume with cubic splines did not improve the relationship.

We read Needhams study with great interest and appreciate the reference. The primary reason Needham might have failed to detect a U-shaped relationship is that their study was not powered to do so. Their sample size was selected to provide 85% power to compare two equally-sized patient groups with a hazard ratio of 0.7. Notably, their study had N=485 as compared to Eichacker’s study N=5,183 which showed the U-shaped relationship observed in our study (N=5,167). 

There were other differences between the studies including Needham’s use of multiple TV/PBWs per patient over time vs. our use of a single TV/PBW per patient, and Needham’s choice of long-term (2-year) mortality as the main outcome variable vs our use of hospital mortality. Needham used Cox proportional hazards modeling vs MLR - of note, all their covariates were chose “a priori” (including CRS – rather than CRS/TBW). We have mentioned and referenced Needham’s study in our discussion section.

9) How is it possible to reconcile the authors’ finding to those from Prevent trial (Simonis JAMA 2018) in which they tested a strategy with low vs intermediate tidal volumes in patients without ARDS?

In our paper, we discuss that current recommendations strongly support the use of low tidal volumes in ARDS and perhaps for all mechanically ventilated patients, but epidemiological studies show that clinicians commonly use intermediate tidal volumes. In this context, our findings and those of the PREVENT trial are complimentary. Both studies failed to show that LTVV as currently recommended is superior to ITVV. Therefore, we referenced the PREVENT trial as supporting our hypothesis. Both papers support the common clinical practice of often employing ITVV for patients instead of strict LTVV. 

Respectfully and appreciatively,

Robert A Raschke MD MS

Corresponding Author.

---

## [Editor Report · Decision Letter 1]

26 Jul 2021

The Relationship of Tidal Volume and Driving Pressure with Mortality in Hypoxic Patients Receiving Mechanical Ventilation.

PONE-D-21-17095R1

Dear Dr. Raschke,

We’re pleased to inform you that your manuscript has been judged scientifically suitable for publication and will be formally accepted for publication once it meets all outstanding technical requirements.

Kind regards,

Corstiaan den Uil

Academic Editor

PLOS ONE
---

## [Editor Report · Acceptance letter]

30 Jul 2021

PONE-D-21-17095R1 

The Relationship of Tidal Volume and Driving Pressure with Mortality in Hypoxic Patients Receiving Mechanical Ventilation. 

Dear Dr. Raschke:

I'm pleased to inform you that your manuscript has been deemed suitable for publication in PLOS ONE. Congratulations! Your manuscript is now with our production department. 

Kind regards, 

on behalf of

Dr. Corstiaan den Uil 

Academic Editor

PLOS ONE